# The Association between Electronegative Low-Density Lipoprotein Cholesterol L5 and Cognitive Functions in Patients with Mild Cognitive Impairment

**DOI:** 10.3390/jpm13020192

**Published:** 2023-01-21

**Authors:** Ping-Song Chou, Sharon Chia-Ju Chen, Chung-Yao Hsu, Li-Min Liou, Chi-Hung Juan, Chiou-Lian Lai

**Affiliations:** 1Department of Neurology, Kaohsiung Medical University Hospital, Kaohsiung Medical University, Kaohsiung 807378, Taiwan; 2Department of Neurology, Faculty of Medicine, College of Medicine, Kaohsiung Medical University, Kaohsiung 807377, Taiwan; 3Graduate Institute of Medicine, College of Medicine, Kaohsiung Medical University, Kaohsiung 807377, Taiwan; 4Neuroscience Research Center, Kaohsiung Medical University, Kaohsiung 807377, Taiwan; 5Department of Medical Imaging and Radiological Sciences, Kaohsiung Medical University, Kaohsiung 807377, Taiwan; 6Department of Medical Research, Kaohsiung Medical University Hospital, Kaohsiung 807378, Taiwan; 7Institute of Cognitive Neuroscience, National Central University, Taoyuan City 320317, Taiwan; 8Cognitive Intelligence and Precision Healthcare Research Center, National Central University, Taoyuan City 320317, Taiwan

**Keywords:** cognitive abilities screening instrument, electronegative low-density lipoprotein cholesterol, L5, mild cognitive impairment

## Abstract

L5, the most electronegative subfraction of low-density lipoprotein cholesterol (LDL-C), may play a role in the pathogenesis of cerebrovascular dysfunction and neurodegeneration. We hypothesized that serum L5 is associated with cognitive impairment and investigated the association between serum L5 levels and cognitive performance in patients with mild cognitive impairment (MCI). This cross-sectional study conducted in Taiwan included 22 patients with MCI and 40 older people with normal cognition (healthy controls). All participants were assessed with the Cognitive Abilities Screening Instrument (CASI) and a CASI-estimated Mini-Mental State Examination (MMSE-CE). We compared the serum total cholesterol (TC), LDL-C, and L5 levels between the MCI and control groups and examined the association between lipid profiles and cognitive performance in these groups. The serum L5 concentration and total CASI scores were significantly negatively correlated in the MCI group. Serum L5% was negatively correlated with MMSE-CE and total CASI scores, particularly in the orientation and language subdomains. No significant correlation between the serum L5 level and cognitive performance was noted in the control group. Conclusions: Serum L5, instead of TC or total LDL-C, could be associated with cognitive impairment through a disease stage-dependent mode that occurs during neurodegeneration.

## 1. Introduction

Mild cognitive impairment (MCI) is an intermediate stage between normal cognitive aging and dementia. MCI refers to cognitive impairment that is more severe than normal aging but does not meet the criteria for dementia. Moreover, MCI is a heterogeneous state. Although all MCI criteria include a quantifiable deficit in cognitive function in at least one domain and the absence of dementia or impaired basic daily function [1], the criteria and subtypes of MCI remain controversial in terms of the aspects of the construct. Furthermore, patients with MCI are three times more likely to develop dementia in 2–5 years compared with age-matched controls [2]. Therefore, identifying biomarkers associated with cognitive performance in patients with MCI is essential.

The brain contains approximately 25% of the cholesterol present in the entire body [3], and cholesterol is critical for maintaining the neuronal myelin membrane, acting as a cofactor for signaling molecules, and developing neuronal plasticity [4]. Furthermore, impaired cholesterol metabolism can affect the modulation of amyloid β (Aβ) and tau hyperphosphorylation, which contribute to neurodegeneration [5].

Decreased cholesterol levels have been reported to promote amyloid precursor protein (APP) processing to a nonamyloidogenic pathway by inhibiting the β-site APP cleaving enzyme 1 (BACE1) and γ-secretase [6,7]. In addition, low cholesterol could decrease the lipid raft in the plasma membrane and then contribute to the localization of ADAM10, which is the major α-secretase enzyme for the proteolytic cleavage of the APP. Cholesterol depletion resulted in the increased α-secretase activity of ADAM10 and the increased nonamyloidogenic α-secretase cleavage of APPs on the cell surface [8]. By contrast, increased cholesterol in the brain tissue could promote BACE1 and γ-secretase activity and increase Aβ42 secretion [9,10]. Both β- and γ-secretases reside in the cholesterol-rich lipid rafts of the plasma membrane, and APPs localizing to the cholesterol-rich membrane domain facilitate β-cleavage and generate Aβ [5,11]. Studies using animal models have reported that high cholesterol could contribute to the upregulation of BACE1 and the downregulation of ADAM10 transcription, which suggests the increased release of Aβ from APP [12].

In addition to the generation of Aβ, studies have suggested that cholesterol may be involved in the aggregation of Aβ and may mediate the neural toxicity of Aβ. Cholesterol modulates lipid rafts by contributing to the formation of GM1 on the membrane, which is the major ganglioside. GM1 binds to Aβ and promotes the aggregation of Aβ in the microdomain enriched with cholesterol [5]. Moreover, increasing membrane cholesterol can induce cytotoxicity and neurodegeneration by activating Aβ-induced calcium signaling [13].

However, previous studies have reported conflicting results on the relationship between blood total cholesterol (TC) levels and cognitive impairment: the effect of cholesterol on cognition is strongly dependent on age [14] and possibly modified by vascular risk factors [15] and cholesterol subfractions [16].

High serum levels of low-density lipoprotein cholesterol (LDL-C) were associated with faster global cognitive decline in older Chinese people without dementia [17]. However, the component of LDL-C that contributes to cognitive impairment remains to be determined. In 1988, a research group led by Avogaro first separated LDL-C into electropositive and electronegative fractions through ion exchange chromatography [18]. In 2003, using anion exchange chromatography, Yang and Chen et al. divided LDL-C into five subfractions, L1–L5, with increasing electronegativity [19,20]. L5, the most electronegative LDL-C subfraction, plays a critical role in the pathogenesis of neuroinflammation and neurodegeneration. L5 causes microglial activation and produces neuroinflammatory stress [21]. Furthermore, L5 induces neurotoxic stress and impairs neuronal differentiation [22]. Therefore, we hypothesized that L5 probably contributes to neurodegenerative disorders by enhancing neuroinflammation and producing neurotoxic effects. However, the evidence for an association between L5 and neurodegenerative diseases is not well-established.

In this study, we hypothesized that L5 contributes to cognitive impairment and examined whether L5 is associated with cognitive performance in patients with MCI.

## 2. Materials and Methods

### 2.1. Participants

This cross-sectional study was conducted in Taiwan. Patients aged >60 years who received a new diagnosis of MCI were enrolled and followed up at neurological clinics at Kaohsiung Medical University Hospital. Age-compatible, healthy older adults were recruited as normal controls from the general population. All participants were required to have a minimum education level of >6 years and be right-handed, as determined by the Edinburgh inventory. All participants underwent a medical history evaluation, physical and neurological examinations, laboratory blood tests, and cognitive assessments.

The diagnosis of MCI by a neurologist was based on the presence of cognitive impairment, as evidenced by an objectively measured decline and a subjective self-report of decline in normal activities of daily living [23]. The exclusion criteria included a history of prior stroke, epilepsy, traumatic brain injury, brain tumors, parkinsonism, major neuropsychiatric disorders, substance abuse, or receiving any cognition-enhancing treatment.

### 2.2. Cognitive Evaluation

A well-trained psychologist administered the Chinese version of the Cognitive Abilities Screening Instrument (CASI), a comprehensive cognitive assessment, to all the participants [24]. The raters of the CASI were blinded to the results of the lipid profiles.

The nine cognitive domains of the CASI include short-term memory, long-term memory, attention, concentration, orientation, constructional praxis, abstraction and judgment, category fluency, and language ability. The maximum score of the CASI is 100, with a high score indicating better cognition. Each participant’s total CASI score and performance across the nine domains was determined. A CASI-estimated Mini-Mental State Examination (MMSE) score (MMSE-CE) was recorded because some of the CASI items could be converted to match the MMSE score [25].

### 2.3. Lipid Profiles and Subfractions of LDL-C

All the participants were asked to fast for at least 8 h before blood sampling and were determined to have no acute illness. Venous blood samples were collected from all participants. Each participant’s LDL-C was extracted according to its density gradient (1.019 to 1.063 g/mL) through sequential potassium bromide density centrifugation, and each LDL-C sample was treated with 5 mmol/L of ethylenediaminetetraacetic acid to prevent it from ex vivo oxidation. Then, the LDL-C samples were separated into subfractions L1–L5 against a salt gradient using the UNO Q12 anion-exchange column (Bio-Rad Laboratories, Berkeley, CA, USA) with the ÄKTA fast protein liquid chromatography system (GE Healthcare Life Sciences, Pittsburgh, PA, USA). Samples with an L1 to L5 subfraction were collected, concentrated, and desterilized according to a method previously described in detail [26].

### 2.4. Apolipoprotein E Genotype

Genomic DNA was extracted from peripheral venous blood leukocytes using QIAamp DNA Blood Kits (Qiagen, Hilden, Germany). Exon 4 of the apolipoprotein E (APOE) gene was assessed using the custom TaqMan probes designed by Applied Biosystems with a real-time PCR system (Applied Biosystems, Waltham, MA, USA) according to the manufacturer’s instructions. DNA fragments were visualized via exposure to ultraviolet illumination (MiniGel1500, Sage Creation, China). Only the patients with 1 or 2 copies of the APOE-ɛ4 allele were considered APOE-ɛ4-positive.

### 2.5. Statistical Analysis

We examined the differences in the demographic data, lipid profiles, L5 concentration, L5 percentage of LDL-C (L5%), and CASI and MMSE-CE scores between the MCI and control groups using Student’s *t*-test, which was followed by an analysis of the homogeneity of variance between groups using Levene’s test for continuous variables and the chi-squared test for categorical variables.

To analyze the association between the electronegative LDL-C and cognitive performance, Pearson’s correlation coefficients were calculated to examine the correlation between L5, L5%, CASI, and MMSE-CE scores. Statistical analysis was performed using SPSS (IBM SPSS 19.0, Illinois, USA). All statistical tests were two-tailed, and an alpha value of 0.05 was considered statistically significant.

### 2.6. Ethical Approval

This study was approved by the Institutional Review Board of Kaohsiung Medical University Hospital (KMUHIRB-20130123) and was conducted in accordance with the Declaration of Helsinki for experiments involving humans. Written informed consent was obtained from all participants.

## 3. Results

The sample size was estimated to be 128 participants in total, split between patients with MCI and people with normal cognition (controls). However, due to the difficulty of analyzing the L5 subfraction, blood samples from a total of 22 patients with MCI and 40 controls were analyzed in this study. The demographic data, cognitive assessment, and lipid profiles of the participants are presented in Table 1. The mean age of the participants in the MCI group was 66.6 ± 7.0 years, and 36.4% of them were men. The mean age of the participants in the control group was 64.6 ± 5.2 years, and 22.5% of them were men. The prevalence of APOE-ɛ4 carriers was 54.5% and 65.0% in the MCI and control groups, respectively. The results of the Student’s *t*-test and the chi-squared test revealed no significant differences in age, sex, the prevalence of APOE-ɛ4 carriers and cardiovascular risk factors between the groups, indicating no sample selection bias in the demographic data between the groups.

The MCI group had a significantly lower education level (10.5 ± 3.6 vs. 13.3 ± 3.6 years, *p* = 0.005), CASI score (86.4 ± 5.0 vs. 93.3 ± 3.0, *p* < 0.001), and MMSE-CE score (25.4 ± 2.6 vs. 28.4 ± 1.2, *p* < 0.001) than the control group. Among the subdomains of the CASI, the MCI group had a significantly worse performance in short-term memory, concentration, abstraction, language ability, and category fluency than the control group (Table 1).

In terms of lipid profiles, no significant differences in TC, HDL-C, LDL-C, or L5 concentrations and L5% were noted between the groups. The post hoc power analysis was 0.457 for comparing L5 concentrations between the MCI and control groups. Using the cut-off value for hyperlipidemia or coronary artery disease [27] of L5 concentrations <1.7 mg/dL and L5% <1.6%, there was no significant difference in the prevalence of individuals with L5 concentrations ≥1.7 mg/dL and L5% ≥1.6% between the groups (Table 2).

In the comparison of demographic characteristics and cognitive performance stratified by the normal range of the L5 concentration and L5% in the control and MCI groups, there was no significant difference in age, sex, education level, MMSE-CE score, or CASI score. In the comparison of the subdomains of the CASI, individuals with L5 ≥ 1.7 mg/dL or L5% ≥ 1.6% in the MCI group had a significantly worse performance in language ability (Table 3). By contrast, L5 ≥ 1.7 mg/dL and L5% ≥ 1.6% were not associated with performance in the subdomains of the CASI in the control group.

No significant correlation was observed between the serum TC and LDL-C and cognitive performance (CASI, MMSE-CE, and subdomains of the CASI) in the MCI and control groups (Table 4). However, a significant negative correlation between the serum L5 concentration and total CASI score was noted in the MCI group (Pearson’s correlation coefficient (r) = −0.431, *p* = 0.045), particularly in the subdomain of language ability (r = −0.438, *p* = 0.042). In the MCI group, serum L5% was negatively correlated with the MMSE-CE score (r = −0.434, *p* = 0.044) and the total CASI score (r = −0.484, *p* = 0.023), particularly in the subdomains of orientation (r = −0.470, *p* = 0.027) and language ability (r = −0.533, *p* = 0.011; Table 4). No significant correlation between the serum L5 concentration and L5% and cognitive performance was observed in the control group.

## 4. Discussion

To the best of our knowledge, this is the first clinical study investigating the association between the electronegative L5 and cognitive function in patients with MCI and older people with normal cognition. The results revealed that L5, but not TC or LDL-C, was associated with impaired global cognitive function, orientation, and language in patients with MCI but not in older people with normal cognition. The range of L5 and L5% levels previously used to predict patients with hyperlipidemia or coronary artery disease could be applicable in predicting the language ability of patients with MCI.

The LDL-C level was reported to be an independent predictor of preclinical cognitive impairment in a study using cognitive event-related potentials [28]. In patients with subjective cognitive decline, high LDL-C levels were associated with cognitive deterioration [29]. A meta-analysis indicated that TC could be a predictive biomarker for Alzheimer’s disease (AD) or cognitive decline in later life, and an elevated LDL-C concentration ( > 121 mg/dL) may be a risk factor for AD, particularly in patients aged 60–70 years old [30,31]. Thus, LDL-C plays a stage-dependent role in the clinical progression from cognitive impairment to AD.

However, community-based cross-sectional studies conducted in China have indicated that a slightly higher LDL-C level is positively correlated with cognitive performance [32] and inversely associated with dementia in people aged ≥50 years old [33]. Furthermore, a prospective follow-up study demonstrated that LDL-C was inversely associated with incident dementia and incident AD in individuals without dementia and vascular risk factors at baseline [15].

A plausible explanation for the conflicting effects of LDL-C on cognition is that the effect of LDL-C on cognition is modified by electronegative LDL-C instead of the effect of TC or LDL-C. Notably, L5, the most electronegative LDL-C fraction in human serum, exhibits atherogenic properties [34]. Not only can atherogenic L5 cause amyloid-mediated platelet activation, platelet aggregation, and hemostasis, but it is also a risk factor for cerebral atherothrombosis [35]. Therefore, L5 contributes to cognitive impairment by causing cerebrovascular dysfunction [36]. In addition, L5 creates neuroinflammatory and neurotoxic stress, impairs neuronal viability and differentiation, and contributes to neurodegenerative diseases [21,22]. Instead of TC or LDL-C, L5 could play a key role in the pathogenesis of cognitive impairment due to neurodegeneration.

Our results support the hypothesis that L5 is involved in cognitive impairment in neurodegeneration. In patients with MCI, serum L5 levels were inversely correlated with cognitive performance, but this relationship was not significant in the controls. Therefore, the stage of progression during neurodegenerative diseases may be related to the effect of L5 on cognitive impairment. In addition, among the cognitive subdomains, serum L5 levels were highly correlated with orientation and language functions in patients with MCI. However, the mechanism of the association between L5 and orientation and language function is currently unclear. Further studies using functional neuroimaging or electroencephalogram analysis could be conducted to evaluate the susceptibility between brain regions in relation to the L5 level.

The study provides new clinical evidence for the association between electronegative L5 and cognitive function during neurodegeneration. However, when the cut-off value for cardiovascular diseases was validated in cognitive impairment, the MCI group was indistinguishable from the control groups. Due to the cuff-off value being developed for evaluating the risk of cardiovascular disease, the inapplicability of the cut-off value for cognitive function could imply that vascular pathology does not contribute to the association between L5 and cognition. On the other hand, according to Wang and colleagues’ previous reports [21,22], L5 probably contributes to cognitive impairment by enhancing neuroinflammation and producing neurotoxic effects. The cut-off value was associated with language ability in the MCI group. As a result, the cut-off value for cardiovascular diseases cannot be used to distinguish between people with normal cognition and those with MCI, but it can be used by clinical physicians to focus on evaluating and treating language function in MCI patients.

Another significant concern raised by our study findings is whether new therapeutic strategies targeting L5 are appropriate for patients with MCI. Treatments with statins for hyperlipidemia [37,38] and with pioglitazone [39], acarbose [40], and insulin [41] for diabetes have been reported to reduce the serum electronegative LDL-C levels through their effects that cause LDL-C modification, decreased systemic inflammation, and glycemic optimization. Therefore, prospective studies with long-term follow-ups are needed to examine the therapeutic potential of L5 in MCI.

This study has some limitations. First, the study had a relatively small sample size. Because the homogeneity of the variance test was not significant, and the demographic data did not significantly differ between the MCI and control groups, the results indicated no bias in the sample selection. However, the inadequate statistical power might have underestimated the effect of L5 on cognitive function. Second, MCI is a heterogeneous state with a multifactorial composition. Although neuroimaging was not conducted for the participants in this study, the MCI diagnoses were determined by a neurologist and were based on clinical symptoms, neurological examinations, laboratory blood tests, and cognitive assessments. Additionally, patients with a history of central nervous system problems and neuropsychiatric disorders were excluded; hence, neurodegenerative MCI was the focus of our study. Third, family histories of metabolic syndromes, cardiovascular diseases, or dementia and the types of medications used for hyperlipidemia and the underlying diseases were not recorded in this study. Due to the fact that there was no significant difference in the prevalence of hyperlipidemia under treatment between the MCI and control groups, the effect of the statin treatment may have been balanced between the MCI and control groups in this study. Fourth, given that the study was cross-sectional in design, the causal relationship between L5 and cognitive impairment could not be determined. To address these issues, additional longitudinal studies are needed to identify and confirm the long-term effects and therapeutic effects of L5 on cognitive impairment in patients with MCI.

## 5. Conclusions

This study demonstrated that serum electronegative L5, instead of TC or total LDL-C, may be associated with cognitive impairment through a disease stage-dependent mode that occurs during neurodegeneration. Furthermore, this study serves as a pilot study for future prospective research with larger sample sizes to investigate whether L5 could be a predictor and a therapeutic target for clinical progression in patients with MCI, particularly in terms of the loss of language abilities.

## Figures and Tables

**Table 1 jpm-13-00192-t001:** Demographic characteristics of the MCI and control groups.

Characteristic	Control (*N* = 40)	MCI (*N* = 22)	*p*-Value
Age, years, mean (±SD) ^1^	64.6 ± 5.2	66.6 ± 7.0	0.209
Sex, man, *n* (%) ^2^	9 (22.5%)	8 (36.4%)	0.242
Education, years, mean (±SD) ^1^	13.3 ± 3.6	10.5 ± 3.6	0.005 **
MMSE-CE, score, mean (±SD) ^1^	28.4 ± 1.2	25.4 ± 2.6	<0.001 **
CASI, score, mean (±SD) ^1^	93.3 ± 3.0	86.4 ± 5.0	<0.001 **
LTM	10.0 ± 0.2	10 ± 0.0	0.160
STM	10.2 ± 1.4	8.0 ± 1.9	<0.001 **
ATT	7.5 ± 0.8	7.3 ± 0.8	0.301
CCT	9.7 ± 0.7	8.4 ± 1.3	<0.001 **
ORI	18.0 ± 0.0	17.6 ± 1.5	0.268
ABS	4.6 ± 1.4	3.8 ± 1.7	0.039 *
JUD	5.1 ± 0.6	5.0 ± 0.5	0.262
LAN	10.0 ± 0.2	9.4 ± 0.9	0.007 **
FLU	8.3 ± 1.7	7.3 ± 1.4	0.010 *
VC	9.9 ± 0.4	9.6 ± 0.7	0.124
APOE-ɛ4, *n* (%) ^2^	26 (65.0%)	12 (54.5%)	0.419
Hypertension, *n* (%) ^2^	14 (35.0%)	9 (40.9%)	0.645
DM, *n* (%)^2^	12 (30.0%)	10 (45.5%)	0.224
Hyperlipidemia, *n* (%) ^2^	4 (10.0%)	2 (9.1%)	1.000
Metabolic syndrome, *n* (%) ^2^	17 (42.5%)	7 (31.8%)	0.409

Abbreviations: ABS—abstraction; APOE—apolipoprotein E; ATT—attention; CASI—Cognitive Abilities Screening Instrument; CCT—concentration; DM—diabetes mellitus; FLU—category fluency; JUD—judgment; LAN—language ability; LTM—long-term memory; MCI—mild cognitive impairment; MMSE-CE—CASI-estimated Mini-Mental Status Examination; ORI—orientation; SD—standard deviation; STM—short-term memory; VC—visual construction. ^1^ Student’s *t*-test; ^2^ chi-squared test; * *p* < 0.05; ** *p* < 0.01.

**Table 2 jpm-13-00192-t002:** Comparisons of lipid profiles, serum L5 concentrations, and L5 percentages between the MCI and control groups.

Characteristic	Control (*N* = 40)	MCI (*N* = 22)	*p*-Value
TC, mg/dL, mean (±SD) ^1^	189.2 ± 33.6	187.4 ± 34.0	0.838
HDL-C, mg/dL, mean (±SD) ^1^	53.4 ± 14.4	51.9 ± 10.0	0.681
LDL-C, mg/dL, mean (±SD) ^1^	116.0 ± 34.8	112.8 ± 29.9	0.714
L5, mg/dL, mean (±SD) ^1^	2.0 ± 1.7	1.5 ± 0.7	0.197
L5 ≥ 1.7 mg/dL, *n* (%) ^2^	19 (47.5%)	8 (36.4%)	0.397
L5%, mean (±SD) ^1^	1.9 ± 1.6	1.5 ± 0.8	0.140
L5% ≥ 1.6%, *n* (%) ^2^	15 (37.5%)	8 (36.4%)	0.929

Abbreviations: HDL-C—high-density lipoprotein cholesterol; LDL-C—low-density lipoprotein cholesterol; MCI—mild cognitive impairment; SD—standard deviation; TC—total cholesterol. ^1^ Student *t*-test; ^2^ chi-squared test.

**Table 3 jpm-13-00192-t003:** Comparisons of cognitive performance stratified by cut-off value of serum L5 concentrations and L5 percentage in the MCI groups.

	MCI					
	L5 ≥ 1.7 mg/dL		*p*-Value	L5% ≥ 1.6%		*p*-Value
	No *N* = 14	Yes *N* = 8		No *N* = 14	Yes *N* = 8	
Age, years, mean (±SD) ^1^	65.3 ± 6.5	68.9 ± 7.7	0.257	66.0 ± 7.6	67.6 ± 6.0	0.613
Sex, man, *n* (%) ^2^	5 (35.7%)	3 (37.5%)	1.000	4 (28.6%)	4 (50.0%)	0.386
Education, years, mean (±SD) ^1^	11.1 ± 3.8	9.4 ± 3.0	0.276	11.6 ± 3.5	8.6 ± 3.0	0.062
MMSE-CE, score, mean (±SD) ^1^	26.0 ± 1.6	24.4 ± 3.6	0.261	26.0 ± 2.0	24.4 ± 3.2	0.159
CASI, score, mean (±SD) ^1^	87.7 ± 3.7	84.0 ± 6.3	0.096	87.5 ± 3.9	84.4 ± 6.3	0.158
LTM	10.0 ± 0.0	10.0 ± 0.0	N/A	10.0 ± 0.0	10.0 ± 0.0	N/A
STM	8.1 ± 2.0	7.8 ± 2.0	0.750	8.4 ± 1.8	7.3 ± 2.1	0.196
ATT	7.4 ± 0.6	7.0 ± 0.9	0.215	7.4 ± 0.6	7.1 ± 1.0	0.508
CCT	8.6 ± 1.2	8.1 ± 1.6	0.451	8.6 ± 1.4	8.1 ± 1.1	0.451
ORI	18.0 ± 0.0	17.0 ± 2.4	0.286	17.9 ± 0.3	17.1 ± 2.5	0.390
ABS	4.0 ± 1.9	3.4 ± 1.3	0.417	3.9 ± 1.9	3.5 ± 1.3	0.579
JUD	5.1 ± 0.5	4.8 ± 0.5	0.139	4.9 ± 0.6	5.0 ± 0.0	0.671
LAN	9.7 ± 0.5	8.9 ± 1.2	0.025 *	9.7 ± 0.5	8.9 ± 1.2	0.025 *
FLU	7.3 ± 1.3	7.3 ± 1.5	0.954	7.0 ± 1.4	7.8 ± 13	0.219
VC	9.6 ± 0.8	9.8 ± 0.7	0.592	9.7 ± 0.6	9.5 ± 0.9	0.519

Abbreviations: ABS—abstraction; APOE—apolipoprotein E; ATT—attention; CASI—Cognitive Abilities Screening Instrument; CCT—concentration; DM—diabetes mellitus; FLU—category fluency; JUD—judgment; LAN—language ability; LTM—long-term memory; MCI—mild cognitive impairment; MMSE-CE—CASI-estimated Mini-Mental Status Examination; ORI—orientation; SD—standard deviation; STM—short-term memory; VC—visual construction; ^1^ Student’s *t*-test; ^2^ Fisher’s exact test; * *p* < 0.05.

**Table 4 jpm-13-00192-t004:** Correlation of serum TC, LDL-C, and L5 level with cognitive performance in patients with MCI.

Pearson Correlation Coefficient	MCI (*N* = 22)							
	TC	*p*-Value ^1^	LDL-C	*p*-Value ^1^	L5	*p*-Value ^1^	L5%	*p*-Value ^1^
MMSE-CE	0.182	0.416	0.137	0.542	−0.409	0.059	−0.434 *	0.044
CASI	0.188	0.403	0.142	0.528	−0.431 *	0.045	−0.484 *	0.023
LTM	N/A	N/A	N/A	N/A	N/A	N/A	N/A	N/A
STM	0.179	0.426	0.169	0.452	−0.290	0.191	−0.325	0.140
ATT	0.058	0.798	0.055	0.807	−0.165	0.463	−0.161	0.474
CCT	0.107	0.637	0.047	0.835	−0.075	0.739	−0.133	0.556
ORI	0.184	0.412	0.175	0.435	−0.418	0.053	−0.470 *	0.027
ABS	0.179	0.426	0.131	0.562	−0.286	0.198	−0.262	0.239
JUD	−0.120	0.595	−0.199	0.374	−0.196	0.383	−0.036	0.872
LAN	0.284	0.200	0.302	0.171	−0.438 *	0.042	−0.533 *	0.011
FLU	−0.257	0.248	−0.286	0.196	0.089	0.695	0.154	0.492
VC	−0.031	0.891	0.003	0.990	0.057	0.802	−0.173	0.440

Abbreviations: ABS—abstraction; ATT—attention; CASI—Cognitive Abilities Screening Instrument; CCT—concentration; FLU—category fluency; JUD—judgment; LAN—language ability; LDL-C—low-density lipoprotein cholesterol; LTM—long-term memory; MCI—mild cognitive impairment; MMSE-CE—CASI-estimated Mini-Mental Status Examination; ORI—orientation; STM—short-term memory; TC—total cholesterol; VC—visual construction; ^1^ Pearson correlation; * *p* < 0.05.

## Data Availability

The data presented in this study are available upon request from the corresponding author.

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
