# Peer review of "The Association between Electronegative Low-Density Lipoprotein Cholesterol L5 and Cognitive Functions in Patients with Mild Cognitive Impairment"

_jpm, 2023, doi:10.3390/jpm13020192_

Round 1

Reviewer 1 Report

I wish to thank the editor for the possibility to review this interesting manuscript. I believe that the manuscript is very well structured and focused on a very relevant topic, of interest for a generic audience.

Please. find below some minor points which, if addressed, I think may significantly improve manuscript's readability and results' dissemination.

1) Do the authors performed any a priori or a posteriori g power analysis on the reached power with this sample size? Please, provide information on this point.

2) Do the authors schedule to conduct a study including a clinical group with dementia? It would be interesting to address this issue ad future development.

3) How do the author interpret the results that L5 level was highly correlated with orientation and language functions? Why L5 should be related to orientation and language function, but not other cognitive functions?

Reviewer 2 Report

In this study, Chou et al report that low-density lipoprotein cholesterol L5 but not LDL-C or total cholesterol (TC) is associated with mild cognitive impairment in elderly people.

The manuscript is well written and conclusive.

Nevertheless, there are some concerns regarding this paper which have to be addressed:

The authors excluded patients with a history of epieplpsy, stroke, traumatic brain injury, brain tumor, or parkinsonism. MCI diagnosis was based on neurological examination, subjective patients’ reports and cognitive tests.

I wonder if the included patients with MCI also underwent imaging, e.g. MRI scans or CT scans to exclude other possible symptomatic causes for cognitive impairment.

A table with concomitant medication of all participants (controls and patients) would be appreciated. 

The authors state that statin therapy might be a possible therapeutic option to reduce L5 – it would be appreciated to state whether the included participants were taking statins at time of study inclusion.

I wonder if there was a family history of metabolic syndromes, cardiovascular diseases, or dementia in the MCI patients?

Furthermore, the authors state that when using L5/L5% cut off levels for cardiovascular diseases, there are no differences between controls and MCI participants. Could you maybe give a little more discussion why you think these cut off values are not appropriate for MCI patients?
